# Interpretable Alzheimer's Disease Classification
# Via a Contrastive Diffusion Autoencoder

Ayodeji Ijishakin [1]   Ahmed Abdulaal [1]   Adamos Hadjivasiliou [1]   Sophie Martin [1]   James Cole [1]

## Abstract

In visual object classification, humans often justify their choices by comparing objects to prototypical examples within that class. We may therefore increase the interpretability of deep learning models by imbuing them with a similar style of reasoning. In this work, we apply this principle by classifying Alzheimer's Disease based on the similarity of images to training examples within the latent space. We use a contrastive loss combined with a diffusion autoencoder backbone, to produce a semantically meaningful latent space, such that neighbouring latents have similar image-level features. We achieve a classification accuracy comparable to black box approaches on a dataset of 2D MRI images, whilst producing human interpretable model explanations. Therefore, this work stands as a contribution to the pertinent development of accurate and interpretable deep learning within medical imaging. Code available at: Contrast-DiffAE.

## 1. Introduction

Deep learning models are increasingly used within the medical domain, which is one of the most safety critical fields (Vilone & Longo, 2020; Marques-Silva & Ignatiev, 2022). As deployment into healthcare expands, greater interpretability of the factors that influence model predictions is essential to ensure safety and trustworthiness (Pawar et al., 2020; Srinivasu et al., 2022; Shaban-Nejad et al., 2021).

Medical imaging has seen a particular influx of deep learning applications in areas including: image classification, segmentation, synthesis, interpolation and denoising (Yousef et al., 2022; Aggarwal et al., 2021; Han, 2021; Haq, 2022).

This is opposed to the rate of clinical adoption of these techniques, which is much smaller. A barrier to closing the gap between research and clinical adoption is the lack of interpretability (Martino & Delmastro, 2022). Consider a radiologist of the future who uses deep learning to help them assess magnetic resonance (MR) images. If the model predicts that a benign tumour will become malignant, but the radiologist strongly disagrees, without an interpretable explanation the output of the model is useless as there is no clear reason to *trust* its prediction.

Beyond healthcare, concerns surrounding the interpretability of AI models have been raised by political bodies such as: the United Nations Educational, Scientific and Cultural Organization (UNESCO), the Organisation for Economic Co-operation and Development (OECD), the government of Australia as well as the United States and the European Union (Marques-Silva & Ignatiev, 2022; UNESCO, 2021; National Science and Technology Council (US), 2019; Gov, 2021; EU, 2021; OECD, 2021). As a response, a plethora of techniques in the field of Interpretable Machine Learning (IML) are being developed and the field can be discretised into two main approaches (Gautam et al., 2021). Post-hoc methods, which interpret the predictions of black-box models after they have made predictions, and self-explainable models (SEMs) whose predictions are accompanied by interpretable explanations.

The literature concerning the former approach is more extensive than the latter (Plumb et al., 2019; Rudin, 2018; Hedström et al., 2022). This is because SEMs have traditionally traded interpretability for accuracy, however recent methodological advancements have increased their predictive capabilities (Plumb et al., 2019; Gautam et al., 2022; Chen et al., 2018; Gautam et al., 2021). As such, we chose to develop an SEM in the present work, to contribute to this comparatively nascent field of IML.

One approach to producing computer vision SEMs is *prototype* learning (Rymarczyk et al., 2021; Kim et al., 2021; Nauta et al., 2020). Prototypes are in-class variants that provide transparency to model predictions because, the final prediction of the model is based on some similarity metric between input data and prototypes (Gautam et al., 2021; Chen et al., 2018). The intuition behind such models is that

---

[1]Centre for Medical Image Computing, Department of Computer Science, University College London. Correspondence to: Ayodeji Ijishakin <ayodeji.ijishakin.21@ucl.ac.uk>.

*Workshop on Interpretable ML in Healthcare at International Conference on Machine Learning (ICML)*, Honolulu, Hawaii, USA. 2023. Copyright 2023 by the author(s).

humans may classify visual objects based on how similar the new visual percept looks to prototypical class examples. For example, a radiologist evaluating a tumour may compare how similar a new MR image looks to prototypical versions of MR images with tumours, and come to a conclusion by weighting the similarities differentially. Prototypical SEMs aim to imbue neural networks with a similar style of reasoning to produce transparent predictions.

Most approaches to prototype learning compare a CNN feature map of an image to image patches drawn from the training set (Chen et al., 2018; Rymarczyk et al., 2021; Nauta et al., 2020; Wang et al., 2021; Kim et al., 2021). In the present work, we extend the prototype learning framework by developing an SEM which predicts the class of an image based on its similarity to training examples (prototypes) within a generative model. Our motivation is the success that generative models have shown at capturing latent semantically meaningful factors (Preechakul et al., 2021; Tomczak, 2022; Mo et al., 2023; Higgins et al., 2016; Dhariwal & Nichol, 2021). These latent factors can be utilised for prototype learning, as they are what define how examples vary within a class, and therefore what the prototypes are. We utilise a new class of generative model, namely the diffusion autoencoder, which has shown promising results in semantic distillation (Preechakul et al., 2021; Li et al., 2023). We combine the diffusion autoencoder with a contrastive loss which brings intra-class images closer in embedding space, whilst pushing away inter-class images. Each prediction from our model is accompanied by a visual explanation, comprised of the nearest prototypes to an image. Thus providing an intuitive explanation for the model's decision.

## 2. Background

### 2.1. Diffusion Models

Below is a brief overview of denoising diffusion probabilistic models (DDPMs), denoising diffusion implicit models (DDIMs) and where diffusion autoencoders fit within this framework.

DDPMs are generative models with both score-based and variational inference interpretations, which use Gaussian diffusion to noise an image to learn a denoising function which captures the data likelihood. The model can be decomposed into two elements: the noising process and the generative process. The noising process maps an input image, $\mathbf{x}_0$, to a standard Gaussian distribution, $\mathbf{x}_T \approx \mathcal{N}(0, \mathbf{I})$, after $T$ successive noising steps. The process is a latent variable markov chain, thus the joint distribution of all latent noisy images, $q(\mathbf{x}_{1:T})$, conditioned on the original input is: $q(\mathbf{x}_{1:T}|\mathbf{x}_0) = \prod_{t=1}^{T} q(\mathbf{x}_t|\mathbf{x}_{t-1})$. At each noising step,

Gaussian noise is added of the form:

$$q(\mathbf{x}_t|\mathbf{x}_{t-1}) = \mathcal{N}(\sqrt{1-\beta_t}\mathbf{x}_{t-1}, \beta_t\mathbf{I}) \tag{1}$$

Where, $\beta_t$, is a hyperparameter which controls the level of noise added and, $\beta_t\mathbf{I}$, is a variance preserving term. After $t$ time steps the noised version of the image, $\mathbf{x}_t$, is also a Gaussian, $q(\mathbf{x}_t|\mathbf{x}_0) = \mathcal{N}(\sqrt{\alpha_t}\mathbf{x}_0, (1-\alpha_t)\mathbf{I})$, where $\alpha_t = \prod_{s=1}^{t}(1-\beta_s)$.

In the generative process the denoising function is modelled, $p(\mathbf{x}_{t-1}|\mathbf{x}_t)$, which maps from our final noise latent, $\mathbf{x}_T$ back to our original data, $\mathbf{x}_0$. To approximate this distribution Ho et al. (2020) introduced learning a function $\epsilon_\theta(\mathbf{x}_t, t)$ which takes as input a time step, $t$ and a noised version of our image at $t$, to predict the noise at that time step using a U-Net. The model is thus trained with a simplified and refactored variational lower bound objective which amounts to an MSE $||\epsilon_\theta(\mathbf{x_t}, t) - \epsilon||$, where $\epsilon$ is the actual noise added at time step $t$ (Preechakul et al., 2021).

DDPMs are very successful at producing high-quality images, but the noise latents $p(\mathbf{x}_{1:T})$ are stochastic and do not capture any semantic information. Song et al. (2021) et al. produced an alternative model by noting that there exists a family of generative models, which share the same objective as DDPMs but have different generative processes. DDIMs are one such generative model which feature the following deterministic (as opposed to stochastic) generative process:

$$\mathbf{x}_{t-1} = \sqrt{\alpha_{t-1}} \left( \frac{\mathbf{x}_t - \sqrt{1-\alpha_t}\epsilon_\theta^t(\mathbf{x}_t)}{\sqrt{\alpha_t}} \right) + \sqrt{1-\alpha_{t-1}}\epsilon_\theta^t(\mathbf{x}_t) \tag{2}$$

#### 2.1.1. DIFFUSION AUTOENCODERS

Preechakul et al. (2021) introduced diffusion autoencoders which extend DDIMs by conditioning the generative process on a semantic latent, $\mathbf{z}_{\text{sem}}$, that is learnt via a separate semantic encoder, $\text{Enc}_{\text{sem}}$. The output of the image encoder is of dimension $d \ll D$ where $\mathbf{x}_0 \in \mathbb{R}^D$. Such that the joint distribution over the generative process is of the form:

$$p(\mathbf{x}_{0:T}|\mathbf{z}_{\text{sem}}) = p(\mathbf{x}_T) \prod_{t=1}^{t=T} p(\mathbf{x}_{t-1}|\mathbf{x}_t, \mathbf{z}_{\text{sem}}) \tag{3}$$

The key idea is that DDIM can be seen as a 'stochastic encoder', $\text{Enc}_{\text{stoch}}(\mathbf{x}_0) = \mathbf{x}_T$, which deterministically maps to the stochastic subcode during the noising process. DDIM is simultaneously the image decoder which takes as input, $\mathbf{z} = (\mathbf{z}_{\text{sem}}, \mathbf{x}_T)$, composed of both the semantic subcode $\mathbf{z}_{\text{sem}}$ and the stochastic subcode $\mathbf{x}_T$. The generative process $p(\mathbf{x}_{0:T}|\mathbf{z}_{\text{sem}})$, the distribution of the image, $p(\mathbf{x}_0)$ and the semantic subcode $\mathbf{z}_{\text{sem}}$ are all learnt simultaneously in an

end to end fashion. The advantage of this training regime is that the semantics are forced into the semantic latent, $\mathbf{z}_{\text{sem}}$, which leads to rich latent representations. These representations can be leveraged for prototype learning, as they separate out the semantics contained within an image, such that we can unpick how these semantics vary within a class. Such rich latent representations are also useful in instances where you have a lack of data (often the case in healthcare), as one may leverage more of the semantic information contained in the image. This further motivates the model class used in the present work as opposed to other generative models (e.g. VAEs or GANs).

## 2.2. Contrastive Learning

Contrastive learning is a prominent approach in self-supervised learning that aims to learn useful representations from often unlabelled data (Chen et al., 2018; Albelwi, 2022; Khac et al., 2020; Khosla et al., 2020). The core idea behind contrastive learning is to optimise a loss function that encourages similar representations for positive pairs (instances of the same class or augmented versions of the same instance) while pushing apart representations of negative pairs (instances from different classes or augmented versions of different instances) (Tian et al., 2020; Liu, 2021). By maximising agreement between positive pairs and minimising agreement between negative pairs, contrastive learning has enabled models to capture meaningful representations, whose similarities reflect similarities at the level of the whole data (Khac et al., 2020; Albelwi, 2022). This regime has demonstrated impressive performance in many computer vision applications, including: aligning text embeddings with image embeddings, image classification with noisy labels, representation disentanglement, and point cloud analysis (Ramesh et al., 2022; Yang et al., 2022; Afham et al., 2022; Li et al., 2022). A particularly successful contrastive loss is SimCLR, proposed by Chen et al. (2018). It takes the following form:

$$\mathcal{L}_{SimCLR}^{(i,j)} = -\log \frac{e^{\text{sim}(z_i, z_j)/\tau}}{\sum_{j=1}^{N} \mathbb{1}_{[k \neq i]} e^{\text{sim}(z_i, z_k)/\tau}} \quad (4)$$

Where $z_i \in \mathbb{R}^d$ is a neural network representation of an image following an image level augmentation (e.g., random crop, flip or noise addition) and $z_j$ is a representation through the same network following an alternative augmentation. Here $\text{sim} : \mathbb{R}^d \rightarrow \mathbb{R}$ is a similarity metric (e.g., Euclidean distance). $N$ is the number of images within a batch, and $\mathbb{1}_{[k \neq i]}$ is an indicator function 1 if $k \neq i$ and 0 otherwise.

As previously outlined, this objective pushes the positive pairs (same image different augmentation) closer together and further way from negative pairs (different images). In

the present work, we adapted this loss to aid the task of interpretable image classification.

## 3. Method

Our model architecture combines diffusion autoencoders with a cosine-similarity based contrastive loss. The model is designed to produce separation within the latent space between classes whilst respecting within class image level similarities. The model processes a batch of images drawn from a dataset, $\mathcal{X} = \left\{\mathbf{x}_{0,i}\right\}_{i=1}^{N}$ where $\mathbf{x}_{0,i} \in \mathbb{R}^D$. Each image has a corresponding label, $\mathcal{Y} = \left\{\mathbf{y}_i\right\}_{i=1}^{N}$, which fit into 2 one-hot encoded classes where $\mathbf{y}_i \in \left\{0,1\right\}^2$. The semantic encoder $\text{Enc}_{\text{sem}}(\mathbf{x}_0) = \mathbf{z}_{\text{sem}}$, maps an image to its semantic subcode, where $\mathbf{z}_{\text{sem}} \in \mathbb{R}^d$. A pairwise similarity metric, $\text{sim}: \mathbb{R}^d \rightarrow \mathbb{R}$ is then computed between $\mathbf{z}_{\text{sem}}$ and all other latents within the batch. The mode class of the $K$ most similar latents is then assigned as the class prediction for $\mathbf{z}_{\text{sem}}$. The similarity metric used is the cosine similarity, defined as:

$$\text{sim}\left(\mathbf{z}_i, \mathbf{z}_j\right) = \frac{\mathbf{z}_i \mathbf{z}_j^T}{\|\mathbf{z}_i\| \|\mathbf{z}_j\|} \quad (5)$$

Following class prediction, the DDIM noising process $q(\mathbf{x}_{1:T}|\mathbf{x}_0)$ maps the image to its stochastic subcode, $\mathbf{x}_T \in \mathbb{R}^D$. Then the DDIM generative process $p(\mathbf{x}_{0:T}|\mathbf{z}_{\text{sem}})$, reconstructs the noised image conditioned on $\mathbf{z}_{\text{sem}}$. Our full model architecture and flow of information can be seen in Figure 1.

Three losses are used to optimize the model. The first loss trains the DDIM model conditioned on the semantic subcode $\mathbf{z}_{\text{sem}}$, it was introduced by Preechakul et al. (2021) and is a modified version of the MSE objective from Ho et al. (2020):

$$\mathcal{L}_{DIFF} = \sum_{t=1}^{T} \mathbb{E}_{\mathbf{x}_0, \epsilon_t} \left[ \|\epsilon_\theta\left(\mathbf{x}_t, t, \mathbf{z}_{\text{sem}}\right) - \epsilon_t\|_2^2 \right] \quad (6)$$

This loss ensures that the generative arm of the model can produce high-fidelity images. It also provides semantically rich latent representations within $\mathbf{z}_{\text{sem}}$, which are useful prototype learning.

We constructed a class-contrastive loss that regularises the embedding space such that images of differing classes are separated, whereas images of the same class are brought together. It is a modified version of the contrastive loss in Chen et al. (2020).

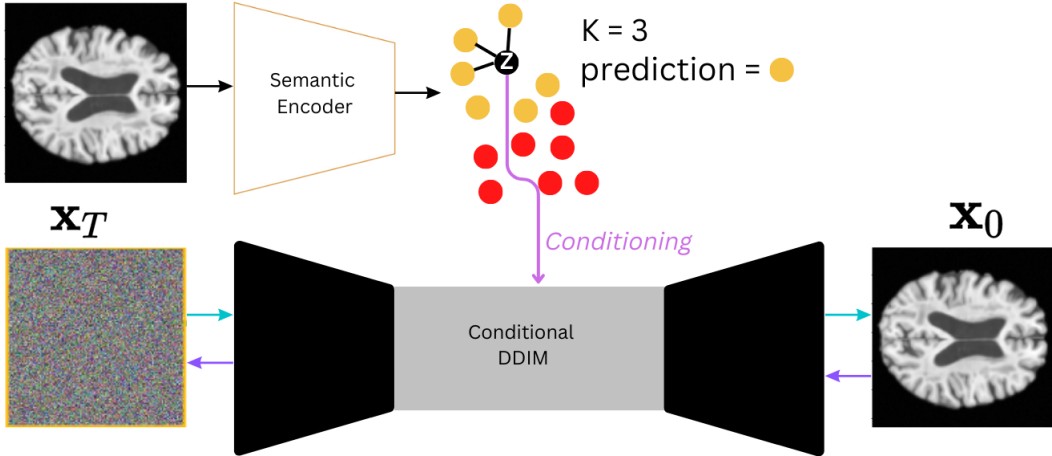

*Figure 1.* Our model pipeline. The image, $\mathbf{x}_0$, is mapped through the semantic encoder to a latent, $\mathbf{z}_{\text{sem}}$ (denoted as $Z$ in the figure for brevity). The latent $\mathbf{z}_{\text{sem}}$ has its similarity to all other latents measured, and the mode class of the $K$ nearest neighbours is used to predict the class of $\mathbf{x}_0$. The noising process then maps $\mathbf{x}_0$, to its 'stochastic subcode' $\mathbf{x}_T$. This is then constructed back to $\mathbf{x}_0$, conditioned on $\mathbf{z}_{\text{sem}}$ through the generative process.

$$\mathcal{L}_{CONTRAST} = \frac{1}{B} \sum_{i=1}^{B} - \log \frac{e^{\text{sim}(\mathbf{Z}^{p,i}, \mathbf{Z}^{p,i})/\tau}}{\sum_{j=1}^{M} e^{\text{sim}(\mathbf{Z}_j^{p,i}, \mathbf{Z}_j^{n,i})/\tau}} \tag{7}$$

Where, $\mathbf{Z}^p$ is a matrix whose rows are latents which are in class 1 and $\mathbf{Z}^n$ is a matrix whose rows are in class 2. Here, $B$ is the number of batches and $M$ is the batch size / 2. Each iteration, $M$ images from class 1 are sampled and $M$ from class 2, making for $2M$ examples. The temperature hyperparameter, $\tau$, controls the magnitude of the loss. The numerator of the above quotient uses the cosine similarity to minimise the distance between examples in the first class in embedding space. The denominator maximises the distance between training examples of class 1 and class 2.

The final loss is the predictive loss:

$$\mathcal{L}_{PRED} = \frac{1}{N} \sum_{i=1}^{N} \mathbf{CE}\left(\hat{\mathbf{y}}_i, \mathbf{y}_i\right) \tag{8}$$

Where, $\hat{\mathbf{y}}$ is the mode of the class labels of the $K$ nearest latents based on the $\text{sim}$ function, $\mathbf{y}$ is the label of an example image and $\mathbf{CE}$, is cross-entropy.

The total objective is the sum of all losses:

$$\mathcal{L}_{TOTAL} = \mathcal{L}_{DIFF} + \mathcal{L}_{CONTRAST} + \mathcal{L}_{PRED} \tag{9}$$

## 4. Experiments

### 4.1. Dataset and Pre-processing

Our dataset was curated for the task of binary classification between Alzheimer's Disease (AD) versus healthy controls (HC). AD is a progressive neurodegenerative pathology which results in widespread brain atrophy and ultimately death (Tatulian, 2022). AD research is increasingly needed due to our globally ageing population, with its global prevalence predicted to rise to 100 million cases by 2050 (Tatulian, 2022). Our dataset consisted of 6137 (AD=1105, HC=5032) 3D structural T1-weighted MR images with isotropic voxel sizes ($1\text{mm}^3$) (mean age=45, range=18-96, std=22.68). Our data were drawn from 10 publicly available datasets. These were: the Australian Imaging, Biomarker & Lifestyle Flagship Study of Ageing (AIBL), the Dallas Lifespan Brain Study, the Nathan Kline Institute Rocklands Sample, the Open Access Series of Imaging Studies-1, the Southwest University Adult Lifespan Dataset, the Alzheimer's Disease Neuroimaging Initiative (ADNI) dataset, the National Alzheimer's Coordinating Center and CamCAN. The images were linearly registered to the MNI 152 brain template using the ANTS package, resampled to $130 \times 130 \times 130$ resolution, n4 bias field corrected using the SimpleITK and skull stripped using the HD-BET package (Avants et al., 2020; Beare et al., 2018; Schell et al., 2019). Following this, 2D medial axial slices were extracted from the 3D images, resized to $64 \times 64$ resolution and normalised to have pixel values between 0 and 1.

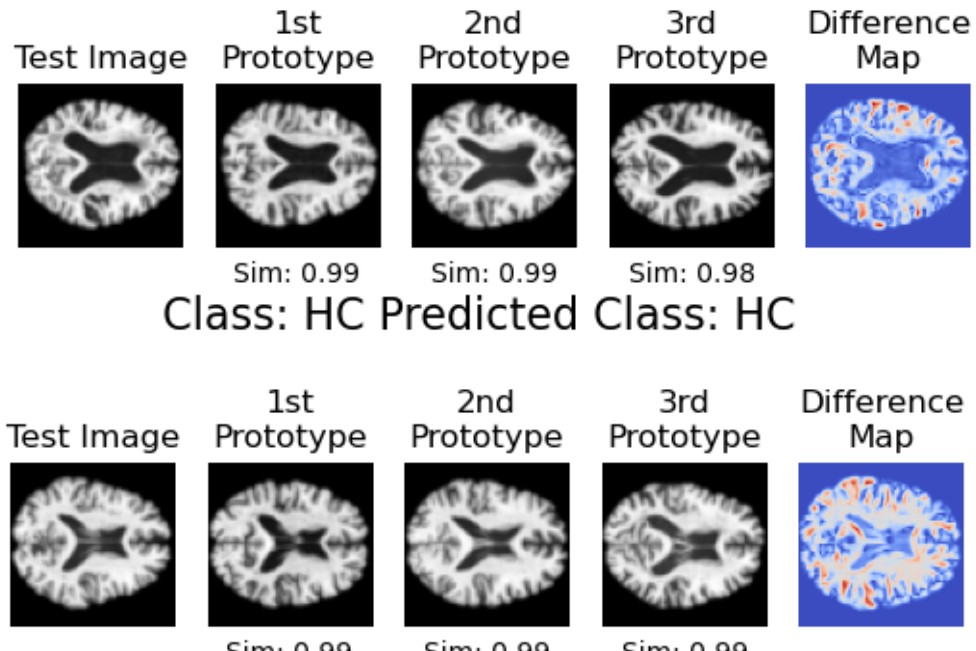

*Figure 2.* Explanations for model predictions on two test examples. On the first row is displayed a test example with AD and on the second row a healthy control. The first, second and third prototypes, are the first, second and third most similar images from the training set. The difference map in the fifth column is displays the image level difference between the first prototype and the test image. The model provides intuitive explanations due to the image level similarity of the classified image to it's prototypes.

### 4.2. Model Design and Hardware

Our diffusion autoencoder model is a downsized version of the U-Net model used in (Preechakul et al., 2021). In the downward path the model expands the input from 1 channel to 32. Following this channel multipliers (2, 4) are applied every 3 convolutional layers two times, making for a channel expansion of $(32 \rightarrow 32 \rightarrow 32) \rightarrow (64 \rightarrow 64 \rightarrow 64) \rightarrow (128 \rightarrow 128 \rightarrow 128)$. The output is then flattened and placed through three attention layers as part of the middle block. The upward path of the U-Net follows the channel expansion of the downward path but in reverse order. Residual connections are used across the two paths where the number of channels are equal. Each convolutional layer is followed by group normalisation and the SiLU activation function. The semantic encoder is the first half of the UNet (no residual connections) plus the middle block. The model comprised 7.6 million parameters in total and was trained on an Nvidia GeForce RTX 4090 graphics card.

### 4.3. Alzheimer's Disease Classification

We first pre-trained the model to reconstruct images without predicting classes with a subsample of the dataset (N=5533, HC=4632, AD=901). This warm-up period lasted for 340

| MODEL | AUTHORS | ACCURACY | INTERPRETABLE |
|---|---|---|---|
| OUR MODEL | - | 0.88 | ✓ |
| CNN | HELALY ET AL. (2021) | **0.97** | × |
| SSAE | MENDOZA-LEON ET AL. (2019) | 0.90 | × |
| XCEPTION | TUFAIL ET AL. (2020) | 0.81 | × |
| CNN + SVM | SETHI ET AL. (2022) | 0.88 | × |
| SVM | UMA-RANI ET AL. (2021) | 0.84 | × |
| TCN | EBRAHIMI ET AL. (2021) | 0.91 | × |

*Table 1.* Comparison of the test accuracy of our approach with other AD vs HC Binary classification studies which use 2D MRI. Both interpretable and black-box models are included and the performant model is in **bold**.

epochs and was based on an evaluation of when the image reconstructions met a high standard according to visual inspection. We then optimised $\mathcal{L}_{CONTRAST}$ and $\mathcal{L}_{PRED}$ for a further 500 epochs with a smaller subsample of the dataset (N=2901, HC=2000, AD=901). We trained with hyperparameter $K = 7$, such that the 7 closest latent representations to a training example were assigned as the class prediction. This was chosen after brief experimentation with $K$, by varying it between [5, 7, 15, 31] and training for up to 10 epochs, where 7 gave the best results.

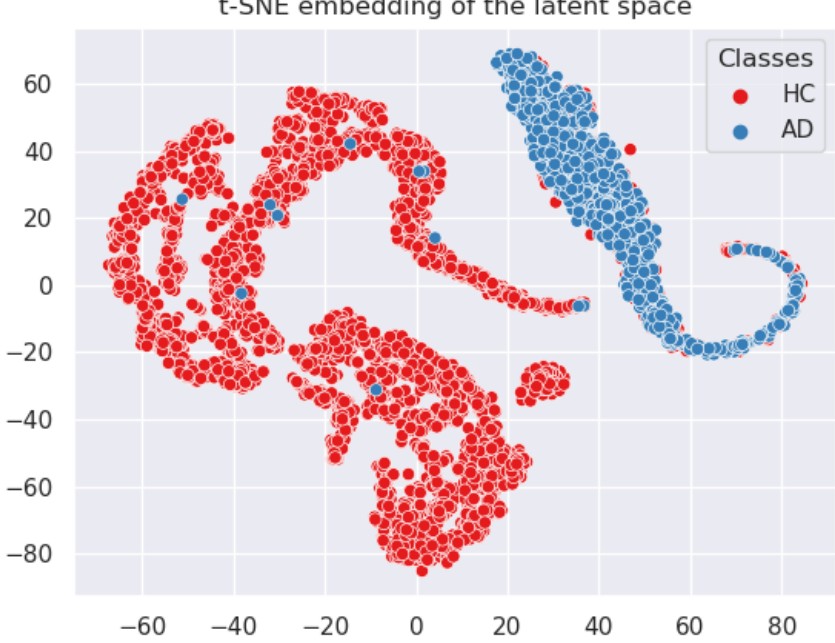

*Figure 3.* A t-SNE embedding of our latent space. The healthy control (HC) subjects are in red and the AD subject's are in blue. There is some entanglement between clusters, likely due to the morphological similarities between the later stages of ageing and neurodegenerative diseases. However, there is broadly clear separation of the two classes, which allows the model to make accurate classifications.

## 5. Results

Table 1 shows the results of our model on a held out test set (N=604, HC=400, AD=204) and how it compares to other approaches to binary classification of AD versus HC via the use of 2D slices drawn from MRI images in other works. We achieve a training accuracy of 95% and a test accuracy of 88%, which is comparable to several contemporary black box models from the literature.

Figure 2 shows two example explanations for model predictions. Here, we treat the $k$th most similar training example as the $k$th prototype. Our results demonstrate that are our predictions are accompanied by interpretable explanations, due to the image level similarity of the prototypes to the examples that they classify (see the Appendix for more examples). Figure 3 shows a t-distributed stochastic neighbour (t-SNE) embedding of the latent space.

## 6. Discussion and Future Work

In this work, we present a model which provides interpretable image classification predictions by mimicking human visual reasoning. Our results show that the accuracy of our approach is comparable to its black-box counterparts on the task of AD classification on 2D MRI slices, whilst

providing intuitive model explanations. Our approach also demonstrates the use of generative models in producing semantic representations, which are useful in the context of prototype learning. This is particularly important due to the use of a diffusion autoencoder, which is a fairly novel generative model. Future work may demonstrate how diffusion autoencoders directly compare to other generative models (e.g VAEs, GANs, and normalising flows) in distilling semantics for prototype learning.

Our t-SNE embedding of the latent space of our model, showed that there is clear separation between the classes of AD versus HC. There exists some entanglement between the classes, as the morphological changes to the brain displayed in older individuals bares a resemblance to neurodegeneration (Cole et al., 2019; De Lange et al., 2020; Cole et al., 2018). As such, it is expected that some of the older individual's in our cohort would (max age = 97) would be clustered close to AD individuals. This may relate to a limitation of the present approach; namely that although it is comparable to black-box approaches, it does not outperform them. This is a problem with SEMs in general but, can be addressed in future extensions of this model class. For example: the contrastive loss could be changed to become more sensitive to the relationship's in the dataset that it is considering. In the present case, that could be the morphological similar-

ity between older people and those with neurodegenerative diseases in general.

There also exists further structure within the latent clusters, which could be used to aid both model predictions and interpretability in future work. This may be achieved by separating out the classes into sub-classes and then defining a metric which can draw distinct prototypes from these sub-classes. Techniques such as graph decomposition, spectral embedding methods as well as classical clustering may be employed to this end. Indeed a recent connection between global spectral embedding methods (e.g. Laplacian Eigenmaps, ISOMAP and Canonical Correlation Analysis) and contrastive self-supervised methods has recently been drawn (Balestriero & LeCun, 2022). Future work may adapt our contrastive loss, in-order to leverage that connection, resulting in prototypes which are derived from a spectral analysis of the latent space. This should increase accuracy because, even if the latent of an older healthy control is closer to an AD patient, they should both be closer to the basis vectors which define their respective classes. In turn, such a model would increase interpretability as a more diverse array of model explanations would be produced (i.e an explanation per basis vector, as opposed to per closest neighbour).

Our model may also be extended by processing the whole 3D image, as the current approach does not utilise otherwise useful information contained across the volume. This is supported by the fact that black-box models of 3D image binary classification of AD also generally perform better than their 2D counterparts (Tufail et al., 2022; Wen et al., 2020; Al-Khuzaie et al., 2021; Mendoza-Leon et al., 2019). Recent advances in DDPM models for 3D medical images may prove useful in extending our model into 3D (Khader et al., 2022). Finally, our prototypes provide global explanations (as the prototypes bare whole image level similarity to classified images), but not local explanations. The use of heat-maps which denote localised regions on prototypes which are useful for classification, may alleviate this problem. In future work, we intend to extend our model to address these limitations, and consider the present work as a preliminary demonstration of this model class.

## 7. Related work

Chen et al. (2018) introduced ProtoPNet, which is a CNN with a penultimate prototype layer prior to the final classification prediction. This approach uses projections of image patches which are prototypical to a class from within the training set and compares them to an embedded input image.

Rymarczyk et al. (2021) extend ProtoPNet by sharing the prototypical image patches between classes, thereby reducing the total number of prototypes required, whilst maintaining accuracy.

Nauta et al. (2020) produced another technique of reducing the amount of prototypes by creating a decision tree based on the similarity of an example image to prototypical image patches.

Wang et al. (2021) constructed an embedding space called TesNet which maps CNN feature maps to output categories. The embedding space is composed of orthonormal basis concepts on a Grassmann manifold. This allows for interpretability as an image is compared to the image patch pairing of the basis concepts when it is classified.

Kim et al. (2021) used a variant of PBPL to study multi-site effects on X-ray image classification.

The above approaches all compare image patches of the final feature maps to image patches from the training set, and we call this approach this patch based prototype learning (PBPL). Although, the PBPL has been applied successfully across computer vision and even in medical imaging, we chose to employ a slightly modified generative model in the present case, to demonstrate that its ability to capture latent semantic factors within the data-distribution may aid prototype learning.

Gautam et al. (2022) contributed ProtoVAE; another approach which uses a generative model for prototype learning. In their approach, prototypes are learnt as an approximation of an orthonormal basis within the latent space of a VAE. However, the VAE backbone creates blurry low resolution images when decoded, which greatly diminishes the interpretability of the model. This is particularly important for medical imaging, where minor details in anatomy or morphology can be strongly related to clinical outcomes. Our approach alleviates this problem by learning the prototypes in the latent space but then using the corresponding whole image from the training set as the model explanation. However, future extensions of our model class may look to high resolution latent prototype learning, as a solution which synthesis both methods.

## 8. Acknowledgements

This work was supported by funding from the Engineering, and Physical Sciences Research Council (EPSRC), the UCL Centre for Doctoral Training in Intelligence, Integrated Imaging in Healthcare (i4health) and the Motor Neuron Disease (MND) Association.

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

# Appendix:

## A. Examples of Model Explanations

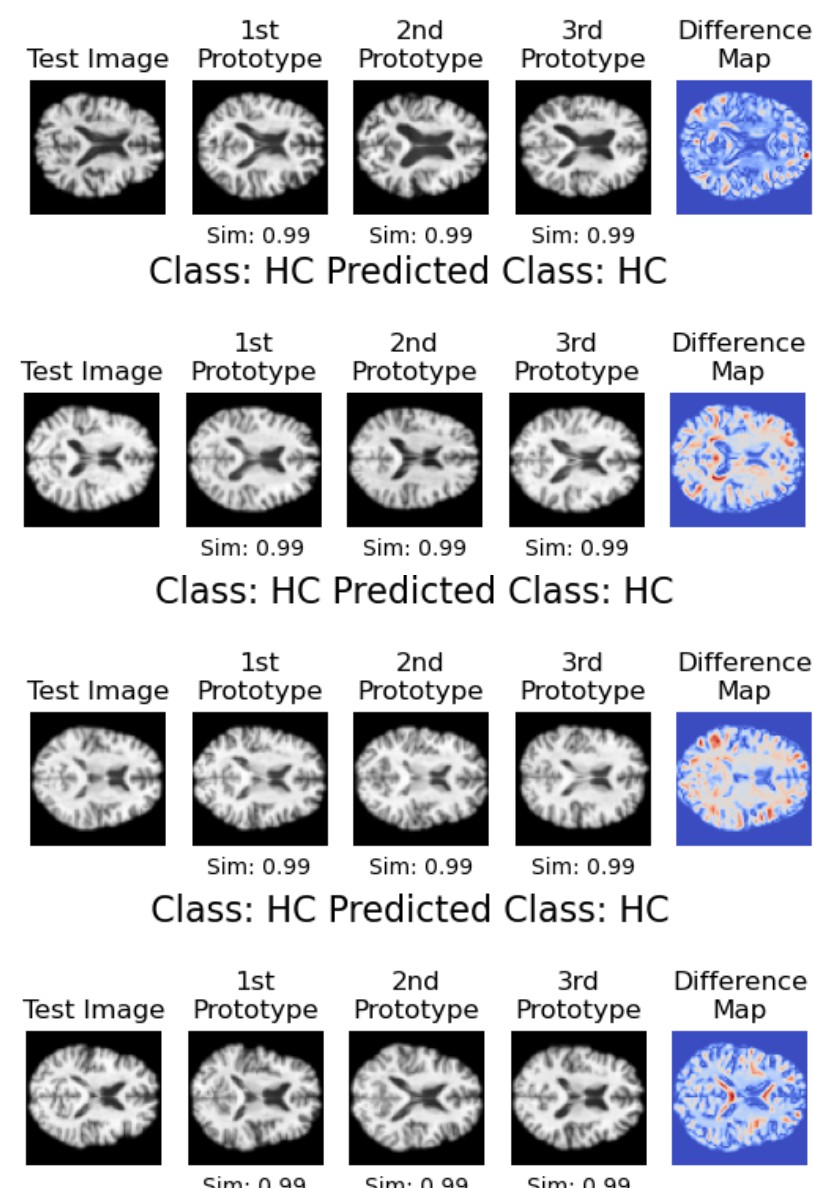

*Figure 4.* Model Explanations on HC test subjects.

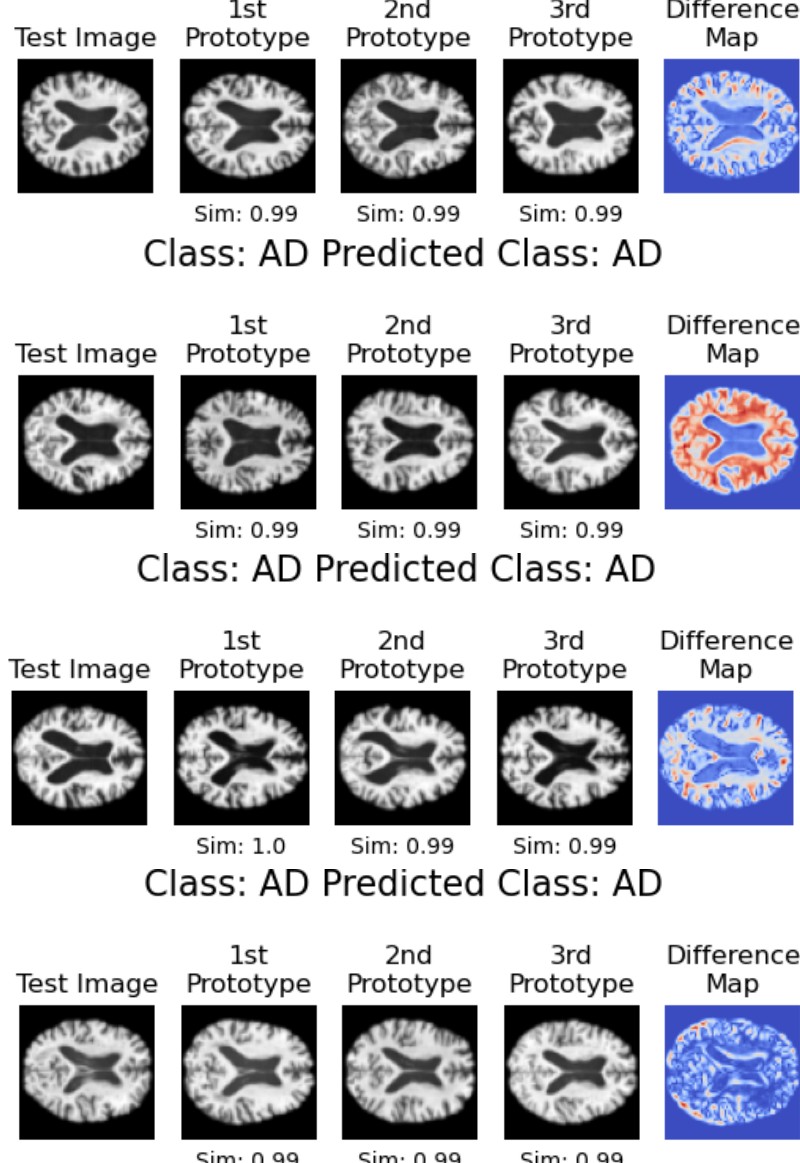

*Figure 5.* Model Explanations on AD test subjects.

# B. Model Reconstructions

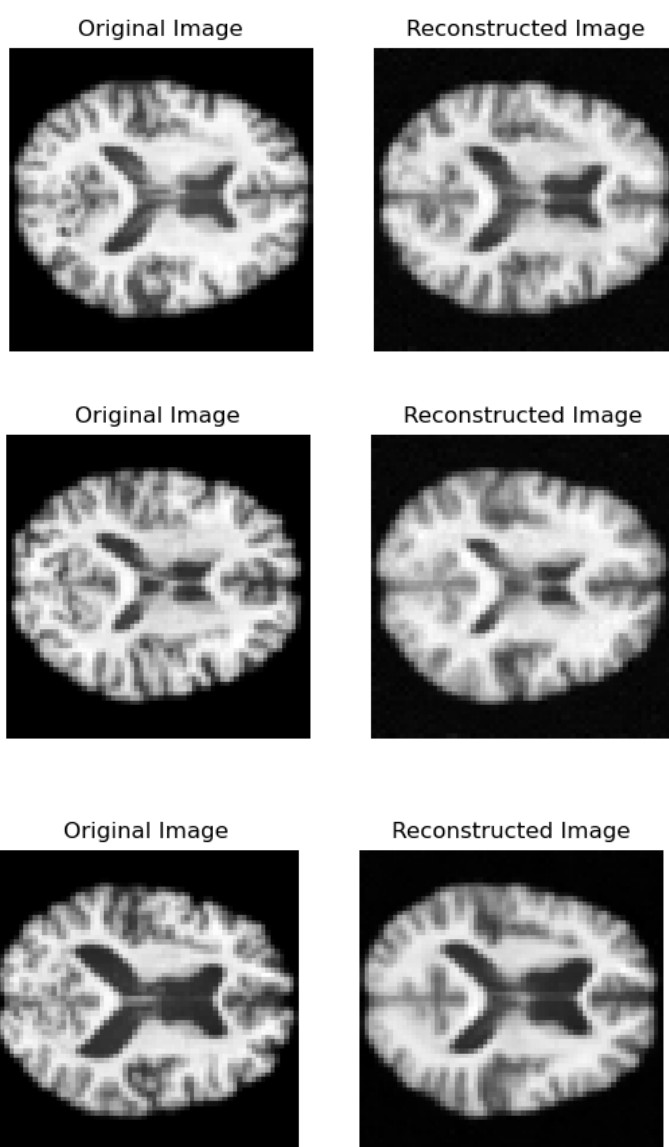

*Figure 6.* Model Reconstructions a).

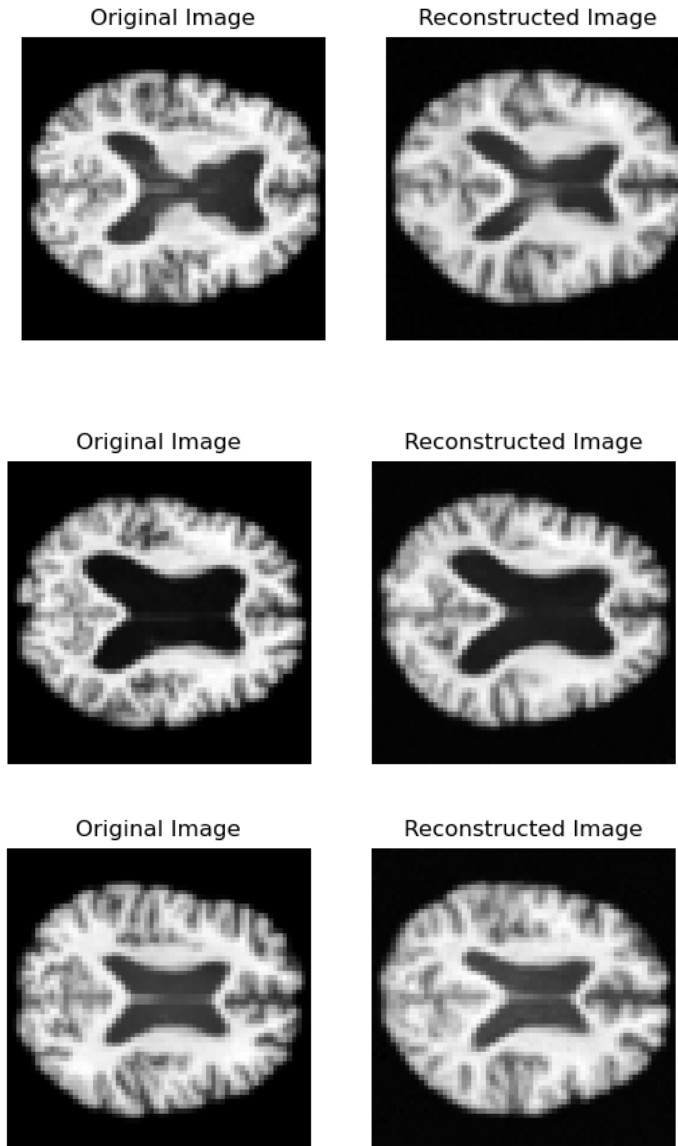

*Figure 7.* Model Reconstructions b).

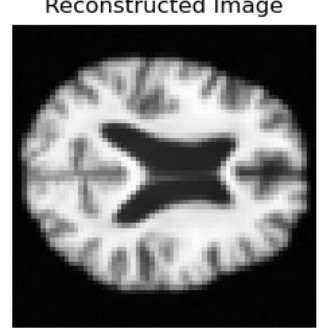

*Figure 8.* Model Reconstructions c).

