# OpenReview forum: "Interpretable Alzheimer’s Disease Classification Via a Contrastive Diffusion Autoencoder. "
_ICML.cc/2023/Workshop/IMLH — IMLH 2023 Poster_

### Official Review · Reviewer_MsVw · 2023-06-07
**This study introduces an interpretable model for image classification that mimics human visual reasoning. Applied to the task of Alzheimer's Disease (AD) classification on 2D MRI slices, the model shows comparable accuracy to its black-box counterparts. The paper also showcases how generative models can be utilized to produce semantic representations, beneficial for prototype learning. Despite its strong performance, the model doesn't outperform black-box approaches, processes only 2D images, and provides only global explanations. The authors suggest addressing these limitations in future work.**

**Rating:** 4
**Confidence:** 4

**Review:**

Pros:

Quality: The research is comprehensive, with a sizable and diverse dataset drawn from 10 publicly available sources. The authors have taken considerable care to ensure the data's uniformity through meticulous preprocessing steps, resulting in a high-quality dataset for the task.

Clarity: The presentation is coherent and well-articulated. The detailed description of the dataset and preprocessing steps provides transparency into the methodology.
The model's ability to mimic human visual reasoning remains a unique aspect, especially in the context of Alzheimer's Disease, a major public health concern.

Interpretability: The model retains its ability to mimic human visual reasoning, providing an intuitive understanding of its predictions.
Diverse and Comprehensive Dataset: The usage of data from multiple diverse sources strengthens the generalizability of the model and its potential applicability to real-world scenarios.

Cons:

Performance: The model does not outperform black-box models, which might limit its attractiveness, despite its interpretability.
2D Image Processing: The model processes only 2D slices of 3D MR images. While this allows for more efficient processing, it may omit potentially relevant information present in the full 3D images.

Lack of Local Explanations: While providing interpretability, the model mainly offers global explanations. Exploring innovative ways to provide local or more granular explanations could have added to its novelty.
Generalization to other Diseases: While the model has been tested on AD and HC data, it's unclear how well it would generalize to other neurological or medical conditions.

Limitation to Structural MRI: The model is designed for structural T1-weighted MR images. Its performance on other types of imaging data (such as T2-weighted or FLAIR images) is not evaluated.

Innovation in Methodology: While the approach of mimicking human visual reasoning is innovative, the underlying methods used for achieving this are built upon existing techniques. The model doesn't introduce a completely new methodology or algorithm.
Novelty might also be limited due to unexplored extensions of the model to other data modalities (e.g., multi-modal data) or other types of diseases, which could have broadened the scope and innovative aspect of the research.

---

### Official Review · Reviewer_HUFW · 2023-06-15
**Well written paper that presents several promising approaches.**

**Rating:** 8
**Confidence:** 3

**Review:**

The authors present an interpretable classification method for Alzheimer's disease based on the similarity of medical images to training examples within the latent space- which they learn using a contrastive diffusion autoencoder. Their method achieves a classification accuracy that is competitive with 2D MRI images classification approaches ,whilst producing human interpretability. Overall the paper, is well written and presents several novel approaches with promising results. In a full paper, the results section can be expanded.

---

### Official Review · Reviewer_jqmE · 2023-06-17
**In this study, a contrastive loss is employed in conjunction with a diffusion auto-encoder backbone. This combination aims to generate a semantically meaningful latent space, wherein neighboring latent representations exhibit similar image-level features.**

**Rating:** 5
**Confidence:** 5

**Review:**

(+) The paper's writing level is satisfactory.

(+) The Background section (section 2) provides readers with ample background information regarding Diffusion Models and Contrastive Learning.

(-) The proposed method lacks novelty to some extent, as it combines diffusion auto-encoders with a cosine-similarity based contrastive loss, which has been previously explored.

(-) The experimental results are limited in scope and may not offer compelling evidence to support the applicability and generalizability of the proposed framework.

(-) While the authors have made an effort to reference relevant studies related to their approach, the comparison between the pros and cons of existing methods and the proposed method is somewhat unclear.

I would recommend that the authors incorporate a more comprehensive technical discussion in the paper. This would provide a clearer and more intuitive justification, thereby enhancing the paper's credibility and confidence. Additionally, improving the figure captions is advisable.

---

### Meta-Review · Area_Chair_1yp1 · 2023-06-20

**Recommendation:** Accept (Poster)
**Confidence:** 5

**Metareview:**

This paper received diverse ratings from reviewers. The strength of this paper is that it used the contrastive diffusion autoencoder that attempts to interpret the Alzheimer disease. However, the Alzheimer's disease itself it's hard to interpret, the paper can be further improved by interpreting a technical method. The results are not very convincing with the proposed method, but it still has a merit of potential interpretability. The authors are recommended to further improve this paper by addressing reviewers' comment.

---

### Decision · Program_Chairs · 2023-06-20

Accept (Poster)